# Detection of Enterovirus D68 in Wastewater Samples from the UK between July and November 2021

**DOI:** 10.3390/v14010143

**Published:** 2022-01-13

**Authors:** Alison Tedcastle, Thomas Wilton, Elaine Pegg, Dimitra Klapsa, Erika Bujaki, Ryan Mate, Martin Fritzsche, Manasi Majumdar, Javier Martin

**Affiliations:** 1Division of Virology, National Institute for Biological Standards and Control (NIBSC), South Mimms, Potters Bar EN6 3QG, UK; alison.tedcastle@nibsc.org (A.T.); thomas.wilton@nibsc.org (T.W.); elaine.pegg@nibsc.org (E.P.); dimitra.klapsa@nibsc.org (D.K.); erika.bujaki@nibsc.org (E.B.); Manasi.Majumdar@nibsc.org (M.M.); 2Division of Analytical and Biological Sciences, National Institute for Biological Standards and Control (NIBSC), South Mimms, Potters Bar EN6 3QG, UK; ryan.mate@nibsc.org (R.M.); martin.fritzsche@nibsc.org (M.F.)

**Keywords:** human enterovirus D68, acute flaccid myelitis (AFM), environmental surveillance, wastewater, next-generation sequencing (NGS), direct detection, COVID-19 pandemic

## Abstract

Infection with enterovirus D68 (EV-D68) has been linked with severe neurological disease such as acute flaccid myelitis (AFM) in recent years. However, active surveillance for EV-D68 is lacking, which makes full assessment of this association difficult. Although a high number of EV-D68 infections were expected in 2020 based on the EV-D68′s known biannual circulation patterns, no apparent increase in EV-D68 detections or AFM cases was observed during 2020. We describe an upsurge of EV-D68 detections in wastewater samples from the United Kingdom between July and November 2021 mirroring the recently reported rise in EV-D68 detections in clinical samples from various European countries. We provide the first publicly available 2021 EV-D68 sequences showing co-circulation of EV-D68 strains from genetic clade D and sub-clade B3 as in previous years. Our results show the value of environmental surveillance (ES) for the early detection of circulating and clinically relevant human viruses. The use of a next-generation sequencing (NGS) approach helped us to estimate the prevalence of EV-D68 viruses among EV strains from other EV serotypes and to detect EV-D68 minor variants. The utility of ES at reducing gaps in virus surveillance for EV-D68 and the possible impact of nonpharmaceutical interventions introduced to control the COVID-19 pandemic on EV-D68 transmission dynamics are discussed.

## 1. Introduction

Infection with enterovirus D68 has been associated with severe respiratory disease in humans in recent years, with increasing evidence of its link with neurological complications causing acute flaccid myelitis (AFM), a polio-like syndrome resulting in long-term or permanent disability in some infected individuals [1,2,3].

EV-D68 was first identified in 1962 but was rarely reported until 2008 when small outbreaks started to emerge in what appears to be mostly a biannual distribution [4,5]. The first known large outbreak of severe respiratory illness associated with EV-D68 infection occurred in the USA between August and December 2014 [6]. Genetically related EV-D68 strains were also found during the same period in Canada, Europe, and Asia, with more than 2000 cases reported in 20 countries [4]. These outbreaks were temporally and geographically associated with an increase in AFM cases, particularly in the USA, and a similar association was observed in Europe, Argentina, and the USA in 2016 and 2018 [1,7]. In particular, 40 AFM cases were reported in England in 2018, with at least 9 of those cases associated with EV-D68 infection [8]. Although EV-D68 circulation in Europe has largely followed a biennial epidemic pattern confined to the autumn season of even-numbered years, there was an unexpected upsurge of EV-D68 infections during the autumn of 2019 leading to 93 reported cases, two with AFM, in five European countries [9]. Despite the above evidence suggesting infection with EV-D68 as a leading cause of AFM, the absence of active and systematic surveillance for EV-D68 means full understanding of this causality is still lacking [10].

As reported for other respiratory and enteric viruses [11,12], the number of EV-D68 detections largely decreased during the COVID-19 pandemic [13,14], which might result from interrupted transmission due to pandemic control measures limiting human physical interactions and increasing body hygiene. However, other reasons such as limited sampling and laboratory testing for EV-D68 might have also contributed to the observed decrease in EV-D68 detections. A recent upsurge of EV-D68 infections, consistent with the known seasonality of this virus typically peaking between September and November, has been reported in several European countries, including the UK [14]. This sudden rise in EV-D68 detections has been attributed to the widespread relaxation of COVID-19 mitigation measures such as travel restrictions, school closures, use of face masks, and physical distancing, as has also been suggested for similar resurgences of other respiratory viruses.

Environmental surveillance (ES) has proven to be a powerful tool for the detection of human viruses of clinical relevance—notably, ES for poliovirus has been used for decades [15], and more recently ES is being used for SARS-CoV-2 with multiple detections across the world during the COVID-19 pandemic [16]. In addition, we and others have used ES to detect and identify non-polio enteroviruses such as EV-D68, for which detection in wastewater samples has been linked to periods of high clinical prevalence [17,18,19].

With this background information in mind, we analysed wastewater samples collected in two locations in the United Kingdom—London and Glasgow—between July and November 2021 for the presence of EV-D68 with an aim to establish genetic links with viruses circulating in previous years. We show the importance of ES for the timely detection of human enteroviruses of clinical relevance and the identification of different EV-D68 genetic variants, as we have shown for SARS-CoV-2 variants of concern throughout the COVID-19 pandemic [20,21,22]. Our results are discussed in the context of the COVID-19 pandemic, as nonpharmaceutical measures introduced to control the pandemic appear to have had a wider impact altering transmission patterns of other respiratory and enteric viruses such as EV-D68.

## 2. Materials and Methods

### 2.1. Wastewater Sample Collection and Processing

Twelve wastewater samples available in the laboratory from our routine poliovirus surveillance activities and collected between July and November 2021 (Table 1 in Results Section 3.1) were selected for this study. One-litre inlet wastewater samples were collected by the grab method or as composite samples during a 24 h period at two sewage plants in Glasgow (Scotland) and London (England), respectively, with a catchment area of approximately 400,000 and 4.0 × 10^6^ people, respectively. Each sample was processed using a filtration-centrifugation method described before and previously validated for the detection of polio and non-polio enteroviruses (including EV-D68) during routine ES for poliovirus as part of our role as a WHO Global Specialized polio network laboratory [23,24]. Briefly, following removal of solids by centrifugation at 3000× *g*, wastewater (70 mL) was filtered through a 0.45 µM Nalgene 500 mL Rapid-flow™ filter (Thermo Fisher Scientific, Waltham, MA, USA) and concentrated down to 400 μL using Centriprep centrifugal filter units with a 10 kDa molecular weight cutoff (Merck Life Science UK Limited, Gillingham, UK) following manufacturer’s instructions. VP1 and whole-capsid EV sequences present in wastewater concentrates were determined by next-generation and Sanger sequencing analysis following methods that have been extensively validated using reference EV strains, laboratory-made EV mixtures, clinical samples, and wastewater concentrates from different locations in comparison with the sequences determined by the Sanger method [25,26,27].

### 2.2. Modified Pan-EV Entire-Capsid Coding Region RT-PCR Amplification (mECRA)

We followed a method designed to amplify the entire capsid sequences of EV strains from all four species A, B, C, and D described before [24,25,26], which involved two independent PCR reactions. Viral RNA was extracted from the wastewater concentrate using Roche High Pure viral RNA kit (Roche Life Science, Mannheim, Germany). RT-PCR fragments covering the entire capsid-coding region and part of the region coding for non-structural proteins (2A–2C) were synthesized from purified viral RNAs by one-step RT-PCR using a SuperScript III One-Step RT-PCR System with Platinum Taq High Fidelity DNA Polymerase (Thermo Fisher Scientific, Waltham, MA, USA). Two different reactions were performed using two primer sets:Primers 5′NCR (5′-TGGCGGAACCGACTACTTTGGGTG-3′) and CRE-R (5′-TCAATACGGTGTTTGCTCTTGAACTG-3′);Primers MM_EV_F2 (5′-CAGCGGAACCGACTACTTT-3′) and MM_EV_R1 (5′-AATACGGCATTTGGACTTGAACTGT-3′).

Reaction conditions were: 50 °C for 30 min, followed by 94 °C for 2 min plus 42 cycles of 94 °C for 15 s, 55 °C for 30 s, and 68 °C for 8 min, with a final extension step of 68 °C for 5 min. Amplified products from both reactions were purified using AMPure XP magnetic beads (Beckman Coulter, Indianapolis, IN, USA). The expected amplicon size for both RT-PCR reactions was approximately 3950 nucleotides (nucleotides 553–4459, relative to PV1 Sabin AY184219 reference strain).

### 2.3. VP1-Nested PCR Amplification and Nucleotide Sequence Analysis of the VP1 Coding Region of EV-D68 Strains by Sanger Sequencing

VP1 nucleotide sequences of the EV-D68 strains identified in the wastewater samples from England and Scotland were analysed by Sanger sequencing. PCR fragments containing VP1 sequences were generated from mECRA products by nested PCR using Dream Taq^TM^ hot start PCR master mix (Thermo Fisher Scientific, Waltham, MA, USA) and three sets of primers:Universal primers: D68_VP1_PubF (5′-ACCATTTACATGCRGCAGAGG-3′) and D68-2ARpub (5′-ACATCTGAYTGCCARTCYAC-3′);Clade B3 primers: D68_VP1_PubF (5′-ACCATTTACATGCRGCAGAGG-3′) and 2016_D68_VP1R (5′-CCTGGACCAGTAGTCACTATATTATG-3′);Clade D primers: 14.001-D68-VP1F (5′-ATTGGACAACTAGAGCATTTACATGAG-3′) and 14.001-D68-VP1R (5′-CAGTATTCACTAACCGAATGTCGTG-3′).

Reaction conditions were: 35 cycles of 94 °C for 15 s, 50 °C for 30 s, and 68 °C for 1 min, with a final extension step of 68 °C for 5 min. Amplified products were purified using QIAquick PCR purification kit (Qiagen, Manchester, UK) and sequenced using an ABI Prism 3130 genetic analyser (Applied Biosystems, Carlsbad, CA, USA).

### 2.4. NGS Analysis of mECRA Products

mECRA products from three representative wastewater samples investigated in this study were analysed by NGS for an additional confirmation of the presence of EV-D68 strains using previously validated and reported methods [24,25,26,27]. Sequencing libraries were constructed through random enzymatic fragmentation using the DNA Prep kit (formerly known as Nextera DNA Flex) and indexed using Nextera DNA Unique Dual Indexes (both Illumina, San Diego, CA, USA). These libraries were pooled in equimolar concentrations according to manufacturer’s instructions and sequenced with 250 bp paired-end reads on MiSeq v2 (500 cycles) kit (Illumina, San Diego, CA, USA). Initial demultiplexing was performed on-board by the MiSeq Reporter software. FASTQ sequencing data were adapter and quality trimmed by Cutadapt v2.10 (30) for a minimum Phred score of Q30, minimal read length of 75 bp, and 0 ambiguous nucleotides. Whole-capsid EV sequences were generated by the de novo assembly of filtered NGS reads as described before [24,25,26,27]. Briefly, filtered NGS reads were assembled de novo using stringent assembly conditions: minimum 50 base overlap, minimum overlap identity of 98%, maximum 2% mismatches per read, and only using paired hits during assembly. In addition, the options to produce scaffolds and ignore words repeated more than 100–1000 times, available in the Geneious 10.2.3 assembler (Biomatters, Auckland, New Zealand), were selected to improve the quality of assembly. Final contig consensus sequences were analysed using the RIVM and BLAST online sequence analysis tools [28,29], and EV serotypes were assigned on the basis of their VP1 sequence identity. EV-D68 VP1 sequences were identified for further phylogenetic studies.

### 2.5. Phylogenetic Analysis of EV-D68 Strains from Wastewater Samples

EV-D68 VP1 Sanger and NGS sequences obtained in this study were compared with those of other EV-D68 strains available in the GenBank database (accessed on 1 December 2021). EV-D68 genome sequences were aligned using the program ClustalW (within Geneious). Molecular Evolutionary Genetics Analysis (MEGA) software package version X [30] was used for phylogenetic analyses. The evolutionary history of aligned sequences was inferred using the neighbour-joining method, with evolutionary distances computed using the maximum composite likelihood substitution method. Sequence divergence was determined by calculating mean pairwise distances.

## 3. Results

### 3.1. Detection of EV-D68 in Wastewater Samples by VP1-Nested PCR and Nucleotide Sequence Analysis

Wastewater samples were processed and analysed as described in Materials and Methods. mECRA and VP1-nested PCR reactions were conducted using RNA extracted from wastewater concentrates followed by Sanger sequencing analysis of positive VP1 PCR products (Table 1).

**Table 1 viruses-14-00143-t001:** Detection of enterovirus D68 in UK wastewater samples by VP1-nested PCR.

		VP1-Nested PCR Positive for Enterovirus D68 (%) ^1^		
Location	Date of Collection	Universal Primers	Clade B3 Primers	Clade D Primers
London	13 July 2021	0	16.7	0
Glasgow	29 July 2021	16.7	83.3	0
London	10 August 2021	33.3	50	0
Glasgow	26 August 2021	83.3	33.3	0
London	14 September 2021	50	50	0
London	28 September 2021	83.3	100	33.3
Glasgow	29 September 2021	66.7	100	0
London	12 October 2021	100	100	50
London	26 October 2021	83.3	100	50
Glasgow	27 October 2021	66.7	100	83.3
London	23 November 2021	100	100	0
Glasgow	24 November 2021	100	100	33.3

^1^ Percentage of RNA replicates from each sample date positive for EV-D68.

Six RNA aliquots per sample date were tested independently. As shown in Table 1, EV-D68 was detected in wastewater samples from both locations from July onwards, with the proportion of positive replicates per sample increasing with time and clade B3 primers generally showing higher sensitivity. This might be because B3 clade-specific primer sequences have higher similarity to the 2021 EV-D68 sequences than those of the universal primers, although sampling effects might also partly explain these results. Clade B3 EV-D68 viruses were detected in all positive samples, and clade D EV-D68 strains were identified in wastewater samples collected in the London sewage plant on 28 September, 12 October, and 26 October 2021 and the Glasgow plant on 27 October and 24 November 2021.

### 3.2. NGS Analysis of EV-D68 VP1 Sequences from Wastewater Samples

Entire-capsid EV sequences were obtained by NGS analysis of mECRA products generated from RNA extracted from two London wastewater samples collected on 28 September 2021 and 12 October 2021 and one wastewater sample collected in Glasgow on 29 September 2021. As shown in Figure 1A, EV strains corresponding to all four species A, B, C, and D were identified with between 11 and 28 different serotypes present in each sample, in line with previous findings of EV serotype prevalence in wastewater samples from both locations, with samples from Glasgow showing lower serotype diversity [25]. This might be due to the fact that samples collected in Glasgow were grab samples, while samples from London were composite samples collected during a 24 h period. In addition, the catchment area covered by the London sewage plant is about 10 times larger than that covered by the plant in Glasgow.

Although some bias might exist in primers amplifying different EV strains better than others, the percentage of NGS reads mapping to different EV serotype sequences gives an indication of their prevalence in the population. EV-D68 was detected in all three samples with prevalence ranging between 4% and 11% of EV mapped reads. Three different EV-D68 contigs corresponding to different genetic clusters were identified in the two samples from London and one in the sample from Scotland (Figure 1B).

NGS results agreed with those from Sanger sequence analysis, with EV-D68 sequences belonging to genetic sub-clade B3 identified in all three samples and clade D EV-D68 viruses found only in the two samples from London. EV-D68 VP1 sequences determined by Sanger and NGS were identical or nearly identical between corresponding samples. However, EV-D68 sequences belonging to two different B3 genetic clusters (named B3a and B3b in this manuscript) were clearly identified by NGS in the two samples from London, with sequence similarity of 95.36–96.3% between them. The minor variant B3b was only identified by NGS and not by Sanger sequence analysis, as this technique can only produce consensus sequence information from the major virus component when more than one closely related strain is present in the same sample and primers match all sequences. As shown in next section, all EV-D68 B3 and D strains from UK 2021 wastewater samples were genetically related to viruses reported mainly in Europe and USA in 2018–2019.

### 3.3. Phylogenetic Analysis of EV-D68 Strains Identified in UK Wastewater Samples

As shown in Figure 2, VP1 EV-D68 sequences from UK wastewater samples analysed in this study were phylogenetically linked to previously reported EV-D68 sequences. B3a viruses were closely related to viruses isolated in 2019 from various European countries. B3b viruses were genetically linked to viruses found in Italy in 2018 showing a characteristic amino acid deletion at amino acid VP1-143 located in structural DE loop thought to have antigenic properties [31]. In addition, clade D EV-D68 wastewater UK strains from 2021 were genetically related to viruses known to have circulated in Europe and China in 2019.

## 4. Discussion

EV-D68 was detected in wastewater samples from London and Glasgow between July and November 2021, before the first clinical cases were reported in the UK in August this year [14,33]. EV-D68 strains belonging to genetic clade D and sub-clade B3 were detected in both locations. Our results demonstrate once more the value of ES for the early detection of clinically relevant viruses circulating in human populations [19,34,35]. ES provides a less biased surveillance system, as it produces data from overall infections, including asymptomatic, rather than clinical data from patients feeling ill, going to a doctor, and having their samples tested. This is highly relevant, particularly considering the absence of consistent and systematic active surveillance for EV-D68 across different countries. Information provided by ES can contribute to a better understanding of EV-D68 transmission dynamics thought to show biannual infection peaks, with likely association with severe disease and AFM cases [4,5] providing a more accurate estimate of overall level of infection in a given population.

Although Sanger sequence analysis of VP1 PCR products from wastewater concentrates was sufficient to identify EV-D68 strains from different genetic clades and can be used to detect other EV serotypes using serotype-specific primers [25], the use of NGS analysis allowed us to rapidly estimate the prevalence of different EV serotypes from all four EV species A, B, C, and D. EV serotype distribution was very similar between all three wastewater samples analysed by NGS, particularly between the two samples from London. Coxsackievirus A6 (CV-A6) was the most prevalent serotype in all three samples, which is in agreement with clinical data showing a prevalence above or close to 50% in clinical samples from September and October 2021 [33]. Other enterovirus A serotypes such as CV-A4 and CV-A5 and enterovirus B serotypes such as echovirus 6 (E-6) and coxsackievirus B2 (CV-B2), as well as EV-D68, were also prevalent in wastewater and clinical samples from the same period. Species C EV serotypes were less prevalent in clinical samples, but EV-C serotypes such as enterovirus A1, A19, and A22 were present in wastewater samples in relevant proportions. EV-C serotypes might be less associated with human disease, as has been suggested in previous studies [25,36]. Enterovirus A71, also linked to severe neurological syndromes in recent years [37], was identified in the London wastewater sample from 12 October 2021. In addition, NGS analysis allowed us to detect the presence of a minor EV-D68 genetic variant (B3b) in wastewater samples from London, not detectable in our Sanger sequencing analysis system. Using a similar approach, we have been able to detect and quantify changes in the prevalence of different SARS-CoV-2 variants throughout the COVID-19 pandemic, closely resembling data from clinical samples [20,21,22].

Interestingly, cocirculation of EV-D68 viruses from sub-clade D and clade B3 in the UK in 2021 was evident from our results, with B3 viruses being more prevalent than D strains, a very similar pattern to that observed in various European countries, including the UK, in 2018 and 2019 [9,19]. The genetic diversity observed among EV-D68 sequences from related genetic clades and clusters suggests rapid and widespread circulation of EV-D68 even during silent detection periods and the existence of important gaps in EV-D68 surveillance worldwide.

The observed recent upsurge of EV-D68 detections in wastewater samples in the UK reported here, mirroring that observed in clinical samples in various European countries, including the UK [14], follows a silent period with no EV-D68 detections reported in 2020 in Europe and USA, where a large number of EV-D68 infections and possible association with AFM cases were expected based on clear biannual patterns observed since 2014 [13,14]. The fact that an increase in EV-D68 infections and/or AFM cases did not occur during 2020 as expected has been attributed to the contingency measures introduced to fight the COVID-19 pandemic by reducing human physical interactions and exposure to infectious agents. The full extent of the interference of these contingency measures on virus transmission patterns in different human populations and long-term future effects remains to be fully assessed, but the exceptional circumstances derived from the ongoing COVID-19 pandemic offer a unique opportunity to better understand human virus transmission dynamics and their impact on associated human diseases.

As a massive wastewater infrastructure has been developed to support SARS-CoV-2 surveillance worldwide, this will not only allow the set-up of additional early alert systems to detect changes in the prevalence of relevant infectious agents known to cause human diseases but also provide invaluable unbiased data for research studies on virus transmission dynamics and, eventually, help in predicting and controlling future epidemics by, e.g., selecting adequate vaccine candidates.

## 5. Conclusions

We detected an upsurge of EV-D68 detections in wastewater samples from the UK between July and November 2021, providing the first publicly available 2021 EV-D68 VP1 sequences, which show co-circulation of genetic clade D and subclade B3 viruses as in previous years. This recent upsurge in EV-D68 detections mirrors that recently described in clinical samples from various European countries and follows a silent period with no EV-D68 detections in 2020. The unexpected absence of EV-D68 in 2020 likely reflects the effect of COVID-19 contingency measures that limited human physical interactions and shows the wider effect of the pandemic on the transmission dynamics of other respiratory viruses and their resurgence following relaxation of pandemic control measures.

## Figures and Tables

**Figure 1 viruses-14-00143-f001:**
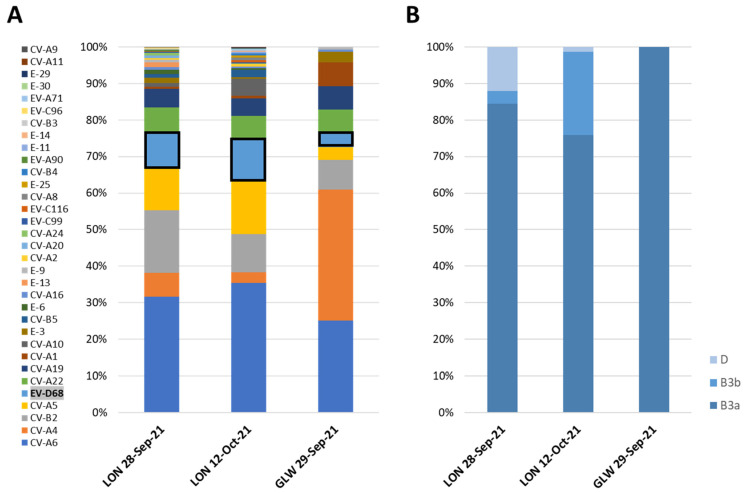
Prevalence of EV serotypes in wastewater samples from the UK estimated by NGS analysis. (**A**) Proportion of NGS reads mapping to capsid sequences of different EV serotypes from all four species A, B, C, and D. Results for EV-D68 serotype are highlighted. (**B**) Proportion of NGS reads mapping to capsid sequences of different EV-D68 genetic variants.

**Figure 2 viruses-14-00143-f002:**
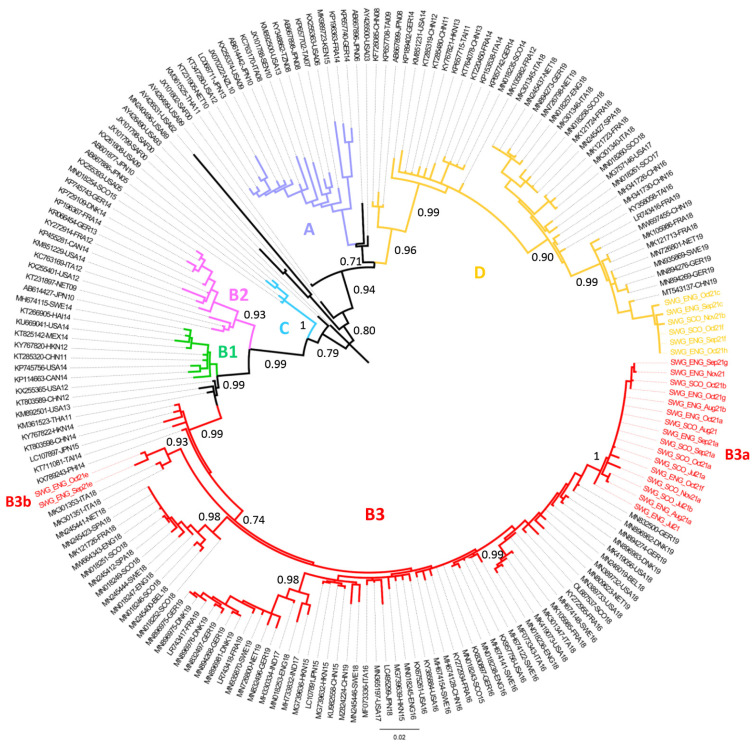
Phylogenetic analysis of VP1 sequences from EV-D68 strains identified in wastewater samples from the UK and representative EV-D68 strains reported worldwide. The evolutionary history was inferred by using the maximum likelihood method and Tamura–Nei model [32]. The tree with the highest log likelihood (−8086.92) is shown. The tree is drawn to scale, with branch lengths measured in the number of substitutions per site. This analysis involved 191 nucleotide sequences. The fractions of replicate trees in which the associated taxa clustered together in the bootstrap test (1000 replicates) are shown next to the branches. VP1 sequences from EV-D68 strains in this study are shown in coloured text. The location of different genetic clades A to D in the tree is indicated. Abbreviations for country names are shown in the Abbreviations section at the end of the manuscript. Evolutionary analyses were conducted in MEGA X [30].

## Data Availability

Nucleotide sequences determined in this study are available from NCBI sequence database with GenBank numbers, OL687538, OL687538, OL687541–OL687553, and OL829826–OL829840 for VP1 sequences and OL829841–OL829847 for entire capsid sequences.

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
