# Peer review of "Detection of Enterovirus D68 in Wastewater Samples from the UK between July and November 2021"

_viruses, 2022, doi:10.3390/v14010143_

Round 1
Reviewer 1 Report
The authors detected an upsurge of EV-D68 detections in wastewater samples from the UK providing the first publicly available 2021 EV-D68. And by sequencing the VP1 sequences, it show co-circulation of genetic clade D and subclade B3 viruses as in previous years. The findings is interesting and important. Some points need to address before publication.
- Why detection of Enterovirus D68 in wastewater samples collected on 13th July by universal primers is negative? and the clade B3 is 16.7%. This need to be discussed.
- The figure 2 is too big and the lable" B3a" cannot be displayed normally.
Author Response
We thank reviewer 1 for the positive comments.
Replies to specific comments:
- Why detection of Enterovirus D68 in wastewater samples collected on 13th July by universal primers is negative? and the clade B3 is 16.7%. This need to be discussed.Reply:
We comment in the text that clade B3-specific primers appear to be more sensitive. This might be because B3 clade-specific primer sequences have higher similarity to the 2021 EV-D68 sequences than those of the universal primers, although sampling effects might also partly explain these results. We have expanded this comment in the text to make it clearer.
2. The figure 2 is too big and the lable" B3a" cannot be displayed normally.
Reply:
We have reduced the size of figure 2 to make label B3a visble
Reviewer 2 Report
Infection with enterovirus D68 (EV-D68) has been linked with severe neurological disease such as acute flaccid myelitis (AFM) in recent years. However, active surveillance for EV-D68 is lacking which makes full assessment of this association difficult. In the present study, the authors used NGS for EV-D68 detections in wastewater samples. This study sounds novel and show the value of environmental surveillance.
Author Response
We thank reviewer 2 for the positive comments